# Glycosylation Analysis of Urinary Peptidome Highlights IGF2 Glycopeptides in Association with CKD

**DOI:** 10.3390/ijms24065402

**Published:** 2023-03-11

**Authors:** Sonnal Lohia, Agnieszka Latosinska, Jerome Zoidakis, Manousos Makridakis, Harald Mischak, Griet Glorieux, Antonia Vlahou, Vera Jankowski

**Affiliations:** 1Center of Systems Biology, Biomedical Research Foundation of the Academy of Athens, 11527 Athens, Greece; 2Institute for Molecular Cardiovascular Research, RWTH Aachen University Hospital, 52074 Aachen, Germany; 3Mosaiques Diagnostics GmbH, 30659 Hannover, Germany; 4Department of Internal Medicine and Pediatrics, Nephrology Division, Ghent University Hospital, 9000 Gent, Belgium

**Keywords:** CKD, eGFR, post translational modifications, PTMs, glycosylation, urine, peptidomics, glycopeptide, IGF2

## Abstract

Chronic kidney disease (CKD) is prevalent in 10% of world’s adult population. The role of protein glycosylation in causal mechanisms of CKD progression is largely unknown. The aim of this study was to identify urinary O-linked glycopeptides in association to CKD for better characterization of CKD molecular manifestations. Urine samples from eight CKD and two healthy subjects were analyzed by CE-MS/MS and glycopeptides were identified by a specific software followed by manual inspection of the spectra. Distribution of the identified glycopeptides and their correlation with Age, eGFR and Albuminuria were evaluated in 3810 existing datasets. In total, 17 O-linked glycopeptides from 7 different proteins were identified, derived primarily from Insulin-like growth factor-II (IGF2). Glycosylation occurred at the surface exposed IGF2 Threonine 96 position. Three glycopeptides (DVStPPTVLPDNFPRYPVGKF, DVStPPTVLPDNFPRYPVG and DVStPPTVLPDNFPRYP) exhibited positive correlation with Age. The IGF2 glycopeptide (tPPTVLPDNFPRYP) showed a strong negative association with eGFR. These results suggest that with aging and deteriorating kidney function, alterations in IGF2 proteoforms take place, which may reflect changes in mature IGF2 protein. Further experiments corroborated this hypothesis as IGF2 increased plasma levels were observed in CKD patients. Protease predictions, considering also available transcriptomics data, suggest activation of cathepsin S with CKD, meriting further investigation.

## 1. Introduction

Chronic Kidney Disease (CKD) is prevalent in 10% of the world’s adult population and is increasingly considered as a silent epidemic, due to its eventual progression to end-stage renal disease (ESRD) [1,2]. CKD is defined by the presence of abnormalities in kidney function and/or structure, for a duration of three or more months. Three aspects: cause, glomerular filtration rate (GFR) and Albuminuria, are important in the classification of CKD [3], with the latter two factors (estimated GFR (eGFR) and levels of Albuminuria) also employed in the diagnosis of CKD [4,5]. However, these parameters have an increased diagnostic value in advanced disease stages, rendering treatment of CKD patients highly challenging. 

Despite its complex composition and unlike other biological fluids, urine is highly stable [6] consisting of proteins and peptides from glomerular filtration, shed epithelial cells (kidney and urinary tract), tubular and seminal secretions, and secreted exosomes [7,8,9]. Urinary proteomics studies have gained momentum in recent years, as they provide opportunities for a non-invasive biopsy. In the case of CKD, urine proteomics have highlighted biomarkers for disease diagnosis and progression; guiding therapeutic strategies [10,11]. 

Protein glycosylation is the most common post translational modification (PTM) [12]. A glycoproteomic study predominantly focuses on the identification of glycoproteins, glycan structures and their sites of attachment in the protein [13,14]. Irrespective of the tremendous advancements in MS/MS-based technologies and respective data analysis solutions, the identification of PTMs, especially glycosylation, continues to be taxing. This can be attributed mainly to the complex and heterogenous nature of glycan structures linked to either Asparagine (N-linked) or Serine/Threonine (O-linked) amino acid residues [15]. The importance of different types of glycosylation (N-linked and O-linked) has already been highlighted in numerous physiological and pathological mechanisms related to inflammation, angiogenesis and cancer [16,17,18]. Along these lines, this study was carried out under the hypothesis that characterization of glycopeptides in the urine can reveal pathological mechanisms in CKD. 

Urinary proteomics to date has identified more than 5000 proteins [19,20,21], while the analysis of urinary glycoproteins is still at its initial stage with the identified number being in the hundreds [22,23]. In the context of CKD, only a handful successful studies on urinary glycopeptidomics exist [24] in their vast majority, focusing on N-linked glycopeptides. Due to the availability of fragmentation knowledge on O-linked glycosylation, an in-depth characterization of O-linked glycans, their attachment sites and protein carriers is warranted. To the best of our knowledge, this is the first study that focuses on identifying “intact” urinary O-linked glycopeptides and exploring their association with CKD. 

Specifically, in this study, urinary glycopeptides were identified by a combination of Capillary Electrophoresis-Tandem Mass Spectrometry (CE-MS/MS) and glycopeptide-specific MS data analysis. The CE-MS technology presents many advantages including high resolution, fast separation, use of inexpensive and robust capillaries, compatibility with common volatile buffers and analytes, and a stable and constant flow, while allowing an easy coupling with Higher-energy C-trap dissociation MS (HCD-MS) for peptide sequencing [25,26,27,28]. The applied approach (summarized in Figure 1) targeted the identification of “intact” naturally occurring urinary glycopeptides in humans, along with information on the glycan structure, composition and attachment sites; with the ultimate aim to identify urinary O-linked glycopeptides in association to CKD and contribute to better characterization of its molecular manifestations.

## 2. Results

### 2.1. Identification of Urinary Glycopeptides 

To identify naturally occurring urinary glycopeptides, urine samples from eight CKD patients and two healthy subjects were analyzed using CE-MS/MS. The MS/MS output datafiles were then processed by Proteome Discoverer 1.4 (the output of protein identifications is provided in Appendix A and peptide identifications in Appendix A), that considered O-linked glycosylation as a variable modification, yielding a list of 120 possible glycopeptides. Further correlation of experimental and theoretically predicted CE migration times based on a regression equation described in detail by Zurbig et al. [29] yielded a shortlist of 65 possible glycopeptides which matched the predicted *m*/*z* and migration time location. The MS/MS spectra outputs of the resulting 65 possible glycopeptide peaks were further manually analyzed, including a comparison of the experimental and theoretical isotopic distributions of the glycopeptide peaks (examples are shown in Figure 2). 

This manual spectral analysis step resulted in a final list of 17 high confidence glycopeptide identifications (Table 1). As shown (Table 1), the 17 glycopeptides were generated from seven different glycoproteins: six out of seventeen glycopeptides were derived from Insulin-like growth factor II [IGF2], three glycopeptides originated from Vitamin K-dependent protein C [PROC], three from Inter-alpha-trypsin inhibitor heavy chain H1 [ITIH1], two from Fibrinogen alpha chain [FGA] and one glycopeptide each of CD99 antigen [CD99], Fibronectin [FN1] and Tumor necrosis factor receptor superfamily member 10D [TNFRSF10D]. The detection frequency and abundance of these glycopeptides was then investigated in the Human Urinary proteome database (*n* = 3810) [21,30,31], as also depicted in Table 1.

### 2.2. IGF2 Glycopeptides in Association to CKD

As the IGF2 glycopeptides were represented at the highest frequency and abundance in comparison to the rest of the identified glycopeptides (Table 1), a more detailed analysis was pursued to investigate their relevance to CKD. Further investigation of the Human urinary proteome database [21,30,31] also revealed three non-glycosylated peptides of IGF2 previously identified (Appendix A). Interestingly, all the detected IGF2 peptides belonged to the E-domain of the protein, as illustrated in Figure 3. 

Four IGF2 glycopeptides were observed in at least 10% of subjects from the Human urinary proteome database (Table 1, first four rows). Given that this high frequency would allow for reliable statistical analyses, these four glycopeptides were considered for further investigation in association with CKD. A correlation of their abundance with clinical parameters (Age, eGFR and Albuminuria) was then performed in the extracted urinary peptide profiles from the database (*n* = 3810), as allowed per data availability. A detailed correlation report is provided in the Supplementary information (Appendix A). As shown (Figure 4a), all peptides exhibited positive correlations with Age, with three of them also exhibiting negative correlations with eGFR based on Spearman’s rank correlation analysis. Associations with Albuminuria could also be occasionally observed suggesting collectively an interplay of all clinical parameters in the determination of peptide abundance.

To better characterize this interplay, multiple linear regression analyses were also conducted (Appendix A), with results being depicted in Figure 4b. As shown, 3 out of 4 glycopeptides (DVStPPTVLPDNFPRYPVGKF; DVStPPTVLPDNFPRYPVG and DVStPPTVLPDNFPRYP) retained an independent association to Age, with DVStPPTVLPDNFPRYPVG also retaining association to Albuminuria. Interestingly one of the peptides (tPPTVLPDNFPRYP) was significantly and independently associated with eGFR [β = −0.014, *p* < 0.0001]. 

To further enhance the validity of this observation related to the specific association of the glycopeptide (tPPTVLPDNFPRYP) with eGFR, correlation analysis of its abundance with eGFR was also performed, following stratification of the available cohort (urinary peptide profiles extracted from the database) in different age groups. In line to the multiple regression analysis, the glycopeptide tPPTVLPDNFPRYP retained a statistically significant and negative correlation with eGFR in five out of six age groups, 18–25 years [*p* < 0.001, *Rho* = −0.69], 41–50 years [*p* < 0.001, *Rho* = −0.43], 51–60 years [*p* < 0.001, *Rho* = −0.56], 61–70 years [*p* < 0.001, *Rho* = −0.56] and 71–100 years [*p* < 0.001, *Rho* = −0.54] with the other 3 glycopeptides showing no significant correlation with eGFR. These results for the peptide tPPTVLPDNFPRYP are also illustrated in Figure 5, where the data were fit using a generalized additive model (GAM), with the detailed correlation test report being provided in Appendix A. 

Given the interesting abundance pattern of the tPPTVLPDNFPRYP glycopeptide, a further analysis was performed targeting its investigation per disease etiology. As summarized in Table 2, the glycopeptide (tPPTVLPDNFPRYP) abundance exhibited statistically significant and negative correlations with eGFR in IgAN, CKD, DKD, Nephroscelrosis, FSGS, tubular nephritis as well as healthy control datasets. For comparison, correlations could occasionally also be observed for two of the other glycopeptides (DVStPPTVLPDNFPRYPVGKF and DVStPPTVLPDNFPRYP), which given the combined results of the Spearman’s correlation test and multiple linear regression analysis (Figure 4), may be related to Age and/or Albuminuria correlations of these peptides, respectively. 

Along the same lines, the urinary IGF2 glycopeptide tPPTVLPDNFPRYP was found at a statistically significant increased abundance in CKD (*n* = 686) urinary peptide profiles in comparison to those of healthy controls (*n* = 229) (Figure 6). 

### 2.3. Increased IGF2 Abundance in Plasma of CKD Patients

The specific association of the urinary glycopeptide tPPTVLPDNFPRYP with eGFR generated the hypothesis that this may be reflective of increased IGF2 levels in CKD. To explore this hypothesis, a pilot study was conducted, targeting quantification of IGF2 protein in plasma. A total of 24 samples of well-matched CKD samples and controls were analyzed corresponding to eGFR < 30 mL/min/1.73 m^2^ (*n* = 12) and eGFR > 90 mL/min/1.73 m^2^ (*n* = 12). Figure 7 shows graphically the results of the enzyme-linked immunosorbent assay (ELISA) analysis supporting a statistically significant increase in plasma levels of the IGF2 protein in the eGFR < 30 mL/min/1.73 m^2^ group when compared with the eGFR > 90 mL/min/1.73 m^2^ group [t = 2.1676, df = 20.294, *p* = 0.04225]. The detailed ELISA analysis report is presented (Appendix A). 

### 2.4. Protease Prediction

To further predict proteases potentially involved in the cleavage of the CKD-specific IGF2 glycopeptide tPPTVLPDNFPRYP (at Serine 95-glycosylated Threonine 96 site), the Proteasix tool was used. The analysis yielded a list of six endopeptidases putatively associated with the cleavage of this peptide at N′ terminal and two endopeptidases responsible for cleavage at C′ terminal (Appendix A). Proteases cleaving at the N′ terminal belonged to the Cathepsin (CTSL, CTSS, CTSK) family, Calpains (CAPN1, CAPN2) and meprin A subunit alpha (MEP1A). Neprilysin protease or membrane metallo-endopeptidase (MME) and matrix metalloproteinase (MMP) were predicted to cleave the C′ terminal of the glycopeptide. Given the specific association of only the IGF2 glycopeptide tPPTVLPDNFPRYP with CKD, predictions that were shared amongst the different IGF2 glycopeptides were disregarded, highlighting Cathepsins, as specifically cleaving glycopeptide tPPTVLPDNFPRYP at the N terminus. Of note, statistically significant increased expression of Cathepsin S (CTSS) in CKD in comparison to Normal Kidney was observed based on transcriptomic datasets as available from the Nephroseq database [*p* = 0.002, *t*-test = 5.797, fold change = 12.099; Appendix A. 

## 3. Discussion

Chronic kidney disease (CKD) is the twelfth most common cause of mortality in the adult population worldwide, with a projection to raise in rank to the fifth position by 2040 [32]. Despite glycosylation being considered a frequent protein modification, its role in CKD is largely unknown. This may be attributed to the highly complex and heterogeneous nature of the glycan structures, especially in the case of O-linked glycosylation, where no evident consensus motif exists. Aiming to fill this gap with this study, naturally occurring “O-linked” urinary glycopeptides were investigated and their association with clinical parameters (Age, eGFR and Albuminuria) relevant to the diagnosis of CKD was established. 

In total, 17 glycopeptides deriving from 7 glycoproteins were identified at high confidence (Table 1). Here, IGF2 derived glycopeptides were observed at the highest frequency and abundance, followed by glycopeptides of PROC, ITIH1 and FGA. The generated glycoforms per protein in the clear majority of cases belonged to consistently similar regions of the protein chain with the different glycopeptides per protein varying amongst each other by one to three amino acids. Along these lines, 5 out of 6 IGF2 glycopeptides exhibited the presence of a glycan structure at the same Threonine 96 position, rendering the latter a highly confident site of glycosylation in the IGF2 protein chain. Similarly, in the case of FGA, a glycosylation site at Serine 609; for PROC, on Threonine 19; and for ITIH1 on Serine 651 could consistently be observed. In the case of IGF2, glycoforms of the same peptide DVSTPPTVLPDNFPRYPVGKF (IGF2) with different glycan positions (S3 and T4) and glycan structures [Hex(1)HexNAc(1)NeuAc(1) for S3 and Hex(1)HexNAc(1)NeuAc(2) for T4] were also observed. The identification of different glycoforms of the same glycopeptide confirms the complex microheterogeneity of O-linked glycosylation and the need for more site-specific glycosylation studies [33]. The narrow CE-migration time (mins) frame observed in between the glycoforms of IGF2 glycoprotein (Table 1), may also be reflective of this O-linked microheterogeneity. The reliable identification of these glycoforms can be associated with the fact that CE-MS/MS recognizes features based on their peptide composition (amino acid sequences) and not the glycan structure. In addition, peptides exhibit an affinity to protons, which further results in identification of [M + *n*H]*^n^*^+^ molecular ions of the glycopeptides [34]. All the glycoforms in this study were identified as singly and/or doubly charged and their isotopic distribution were compared to the theoretical distribution for validation. 

Analysis of O-linked “intact” glycopeptides, has also been conducted by Belczacka et al. and Halim et al. [23,35]. In an analogous manner, these authors also focused on identification of intact urinary glycopeptides (both O-linked and N-linked) in healthy [23] or in association to cancer [35]. Some of the glycosylation sites highlighted in our study have also been reported in these and other studies: Threonine 19 in PROC protein was also identified as a highly confident O-linked glycosylation site in Belczacka et al. and Halim et al. [23,35]. Along the same lines, Darula et al. [36] and Campos et al. [37] detected CD99 as an O-linked glycoprotein with Threonine 41 as the glycan attachment site, a finding which was also supported in our study. N- and O-linked glycosylation are widely stated in the literature for FN1 [38] and TNFRSF10D [37] proteins; nevertheless, no previous reports for the presently shown O-linked glycosylation sites at Threonine 19 and Threonine 69, respectively, exist. 

The IGF2 peptides exhibited high frequency, high abundance, and predominance in comparison to glycopeptides originating from other urinary proteins, which allowed for their further statistical analysis. Translated IGF2 is produced in the endoplasmic reticulum, in the pre-pro-protein form consisting of a 180 amino acid chain, divided into 6 domains (A to E) plus a 24 amino acid signaling peptide. In the post-translational process of pre-pro- IGF2, the E-domain undergoes O-linked glycosylation in 12 plausible sites, by addition of N-acetyl galactosamine residues, and in the trans-Golgi compartment sialic acid side chains are added to N-acetyl galactosamine. Glycosylation stimulates the next step in the processing of pro-IGF2, where the E-domain is cleaved by PCSK proteases, yielding the “mature IGF2” consisting of 67 amino acids and a molecular weight of 7.5 kDa. This (mature IGF2) is then released in the blood stream by exocytosis, where its pleiotropic roles associated with growth and development are carried out in an auto-/para-/endocrine mode [39,40,41,42]. 

Based on the abovementioned post-translational processing, numerous proteo-forms of IGF2, differing in size and/or glycosylation pattern exist, of varied affinity to the same targets (IGF2 or IGF1 receptors, insulin receptors). In a healthy adult, IGF2 (or “mature” IGF2) devoid of any glycosylation is easily degradable in the bloodstream [43]. Several IGF binding proteins (IGFBPs) exist in the extracellular matrix, forming IGF2 binary or ternary structures (IGF2-IGFBPs-acid labile subunit (ALS)) to increase stability of the protein and regulate its concentration in the bloodstream, prior to its degradation that occurs upon binding with the IGF2 receptor (IGF2R). O-linked glycosylation has been linked to inhibition in the formation of such ternary protein complexes, resulting in increased affinity for IGF1R, IR-A and IR-B [44,45]. Collectively, it is suggested that the “mature IGF2” protein form, important for maintaining homeostasis in adults, is most likely present in inactive ternary complexes, while detectable IGF2 proteins are in their vast majority proteo-forms of IGF2 consisting of O-linked glycosylation [46,47]. Along these lines in our study, all identified IGF2 peptides belonged to E-domain of IGF2, i.e., 92–180 aa, outside the “mature IGF2” (25–91 aa). Of note, and in agreement to the abovementioned evidence, in the database, only three unglycosylated urinary peptides of IGF2 could be identified, which were observed at significantly lower abundance and frequency in comparison to the glycosylated forms (Appendix A). 

Correlation analysis indicated moderate to strong [*Rho* = +0.27 (mean)], positive and statistically significant [*p* < 0.0001] correlation of all four IGF2 glycosylated peptides with Age with three of the glycopeptides (DVStPPTVLPDNFPRYPVGKF, DVStPPTVLPDNFPRYPVG and DVStPPTVLPDNFPRYP) maintaining this association after multiple linear regression analysis. This result enhances the biological relevance of IGF2 glycoforms and a need for their individual study. Importantly, one of the peptides (tPPTVLPDNFPRYP), showed strong association with eGFR independent of Age and Albuminuria, as supported by all applied analyses (Spearman’s rank correlation, multiple linear regression analysis and Wilcoxon rank sum test). Collectively, this finding indicates that with aging and deteriorating kidney function, alterations in IGF2 proteoforms take place, which may be reflective of respective changes in the mature IGF2 protein abundance and function as well as associated protease activity. Our results corroborate to some extent this hypothesis, as increased IGF2 plasma levels were observed in CKD patients versus the controls. In addition, the protease predictions, also combined with available transcriptomics data, suggest an activation of cathepsin S with CKD, meriting further investigation through future studies. 

Increased levels of IGF2 in comparison to controls, in multiple diseases including cancer and diabetes have been detected [47]. In brief, associations of increased serum levels of IGF2 with hypoglycaemia [48] as well as of total and free IGF2 (defined as bound or unbound fractions of IGF2 with IGFBPs, respectively) with obesity accompanying Type 2 Diabetes mellitus (T2DM) have been suggested [49]. The E-domain of the IGF2 protein, also known as preptin has been further associated with obesity [50] and T2DM [51]. Interestingly, the upregulation of IGF2 in diabetic nephropathy (DKD) has also been proposed following analysis of renal biopsies from normal (*n* = 9) and early T2DM patients with (*n* = 9) or without (*n* = 11) histopathological characteristics of DKD [52]. In our study, associations to different disease etiologies including DKD could be observed for the tPPTVLPDNFPRYP glycopeptide. As suggested [53], any disturbance occurring in the somatotropic growth hormone axis can contribute to causal mechanisms in CKD. Interestingly, Fan et al. [54] recently reported a metabolic pathway that may also explain the increase of IGF2 in CKD. The study highlighted the important role of an endoplasmic reticulum protein RTN3 in the IGF2-JAK2-STAT3 pathway, where the reduction in RTN3 in the kidneys results in increased transcription of IGF2, which in turn activates the JAK2-STAT3 pathway, ultimately inducing kidney fibrosis and CKD. This pathway was proposed using animal (mice) models and cell lines, hence its translatability to humans is pending.

In summary, with this study naturally occurring “intact” urinary glycopeptides were identified highlighting IGF2 O-glycosylated proteoforms correlating with pathophysiological parameters, such as aging and kidney function, and apparently reflecting at least in part changes in IGF2 levels. The ability to probe these features simultaneously, i.e., site occupancy and O-glycan microheterogeneity, offers a unique opportunity towards untangling and identifying isoform-associated unique functions (similar to the observed specific association of the IGF2 glycopeptide tPPTVLPDNFPRYP with CKD). The results thus open up multiple questions meriting further investigation, including understanding the mechanism of this IGF2 peptide specific association and likely proteolytic cleavage by Cathepsins and their overall impact on CKD progression. 

## 4. Materials and Methods

### 4.1. Study Population–Urine Samples

For identification of naturally occurring intact urinary glycopeptides, urine samples from eight CKD patients and two healthy volunteers were analyzed by CE-MS/MS, under ethics-approved protocols (Hannover Medical School, Ref. No. 3115-2016). The respective clinical information is presented in Table 3. 

For the association of the identified glycopeptides with Age, eGFR and Albuminuria, anonymized CE-MS urinary peptide profiles from the Human Urinary Proteome database [21,30,31] were considered. These peptide datasets generated in previous studies [55,56,57,58,59,60,61] were selected based on availability of information on their diagnosis (control or CKD etiology), Age, eGFR and Albuminuria values. The distribution of disease etiology for the retrieved 3810 datasets is presented in Table 4. 

### 4.2. Sample Preparation

Standard operating procedures (SOPs) were followed for urine sample collection, storage and further processing, as described in previous publications [62,63,64]. Briefly, 700 µL of an aqueous solution consisting of 2 M urea, 10 mM NH_4_OH and 0.02% sodium dodecyl sulfate (SDS) was used to dilute 700 µL of the thawed urine sample. To discard high molecular weight proteins, the sample was subjected to ultrafiltration using a Centristat (20 kDa molecular mass cut-off) centrifugal device (Sartorius, Göttingen, Germany) and centrifugation speed was set to 3000 rcf, to obtain approximately 1.1 mL filtrate. To eliminate urea, salts and electrolytes from the filtrate, a desalting process was carried out utilizing a PD-10 column gel (GE Healthcare Bio Sciences, Uppsala, Sweden) which was pre-equilibrated with 0.01% NH_4_OH in HPLC-grade water. The processed urine samples were then lyophilized and stored at 4 °C. Prior to the CE-MS/MS analysis, the samples were reconstituted in 10 µL of HPLC-grade water. 

### 4.3. CE-MS/MS Analysis

The CE-MS/MS instrumental set up comprised a P/ACE MDQ capillary electrophoresis system (Beckman Coulter, Fullerton, CA, USA) connected to an Orbitrap Velos Fourier Transform (FT) MS (Thermo Finnigan, Bremen, Germany). The protocol for CE-MS/MS analysis and peptide sequencing based on data-dependent high (40%) energy collision dissociation (HCD) for the top 20 ions, has been previously described [65]. Briefly, Proxeon nano spray fitted with Agilent ESI sprayer (operated in positive ion mode) was utilized to ionize 230 µL-aliquot (1:50) of the processed urine sample. The ionization occurred at a voltage of 3.4 kV with 275 °C as the capillary temperature. Operating conditions were set to MS/MS mode that scanned from 350 to 1500 amu and triggered sequencing at a set threshold of 5000 counts. Ion resolution was 60,000 for MS1 and 7500 for MS2, with a detection limit of 0.05–0.2 fmol. Finally, full-scan MS spectra were acquired in the range of 300–2000 *m*/*z*, depicting the sequentially isolated fragmentation ions. The CE-MS/MS proteomics data have been deposited to the ProteomeXchange Consortium via the PRIDE partner repository with the dataset identifier PXD039829 and 10.6019/PXD039829.

### 4.4. Glycopeptide Analysis

Identification of glycopeptides was carried out using Proteome Discoverer 1.4 software. The SEQUEST search engine was applied for analysis of the CE-MS/MS raw files against the entire reviewed non-redundant human database downloaded as FASTA from UniProt (May 2022). The search parameters were set to no enzyme, HCD as the activation type, precursor mass tolerance at 10 ppm and fragment mass tolerance at 0.02 Da. In addition, only 2 maximum missed cleavage sites were allowed along with the peptide length set to 6–144 amino acids. There were no fixed modifications selected, while the variable modifications included oxidation of proline (+15.995 Da), Hex(1)HexNAc(1)NeuAc(1) of serine, threonine (+656.228 Da) and Hex(1)HexNAc(1)NeuAc(2) of serine, threonine (+947.323 Da). The latter two accounts for the most commonly occurring structures of O-linked glycans [66,67,68]. Three dynamic modifications per peptide and a false discovery rate of 1% for peptide identifications were allowed. To eliminate false-positive associations, a selection process based upon correlation between the experimental and theoretically predicted CE-migration time at the working pH of 2.2 of the identified peptides was applied, as described previously [29]. Each of the MS/MS fragment spectra of the resulting glycopeptide peaks were further analyzed manually, including a comparison of the theoretical and observed isotopic distributions.

### 4.5. Study Population–Plasma Samples 

Plasma samples from CKD patients from the registered Biobank Kidney Ghent (BB190096) at the Ghent University Hospital in Belgium were analyzed under the ethics-approved protocol number (EC2010/033; Ghent University Hospital Ethics Committee). Two well-matched groups of 12 samples each were formed corresponding to eGFR > 90 mL/min/1.73 m^2^ and CKD patients with eGFR < 30 mL/min/1.73 m^2^, as shown in Table 5. 

EDTA plasma samples were collected under SOPs. In brief, blood samples were centrifuged at 2095× *g* for 10 min at room temperature, within 30 min after collection. Samples were then aliquoted into labelled cryovials and stored in an upright position at −80 °C. 

### 4.6. ELISA Assay

The quantification of Insulin-like growth factor-II (IGF2) protein (7.5 kDa) in plasma samples was performed by enzyme-linked immunosorbent assay (ELISA) using the Quantikine Human IGF-2 ELISA Kit (#DG200, R&D Systems, Minneapolis, MN, USA) as per the manufacturer’s instructions using 10 μL of plasma. 

### 4.7. Statistical Analysis

Statistical analysis and results presented in this manuscript are based on R programming (R version 3.6.0 with IDE: R Studio Version 1.2.5, Boston, MA, USA). A peptide frequency threshold of 10% was applied, as a pre-requisite for statistical analysis. The peptide intensity values were Log10 transformed and entries with missing/no values were eliminated. Spearman’s rank correlation test and multiple linear regression analysis were used to define associations between abundance of the identified glycopeptides and clinical parameters (Age, eGFR and Albuminuria). In R, a correlation matrix was created with rcorr function of Hmisc package, while the regression analyses were performed with cor.test and lm functions of stats package. Matching for age and sex was performed in R using the MatchIt function where a 1:1 ratio was selected with the ‘nearest neighbor’ method. Wilcoxon Rank Sum test was applied to compare the glycopeptide intensities between groups (healthy subjects, *n* = 229 and CKD patients, *n* = 686). Welch two sample *t*-test was applied to compare the IGF2 protein levels between the two eGFR groups (*n* = 12 per group of plasma samples). Box and Whisker plots were generated in R using the ggplot2 package and colors in the correlation plot developed using a generalized additive model (GAM) were depicted with the viridis package. 

### 4.8. Proteasix Analysis

An open-source tool “Proteasix” (http://www.proteasix.org, accessed on 3 October 2022) [69,70] was used to predict proteases involved in the cleavage of the identified glycopeptides. Briefly, Proteasix exploits information from databases such as UniProt Knowledgebase, MEROPs, CutDB and literature. In this study, we employed the “Predicted” tool of Proteasix for increased coverage in proteolysis information, using the default search parameters. 

### 4.9. NephroSeq Analysis

The transcriptomics data analysis tool Nephroseq v4 (www.nephroseq.org, accessed on 23 December 2022) was utilized to investigate the expression of predicted proteases in existing human transcriptomics datasets. Initially, CKD datasets were identified using the primary filter: Group > chronic kidney disease. The mRNA expression levels of proteases in CKD vs. Normal Kidney were then searched in the available datasets. Significance was defined as *p* < 0.05 and with at least a 1.5-fold change in the expression levels. 

## Figures and Tables

**Figure 1 ijms-24-05402-f001:**
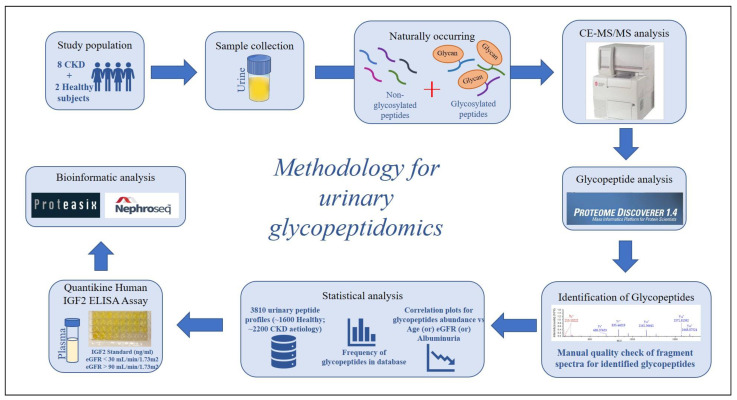
Study workflow.

**Figure 2 ijms-24-05402-f002:**
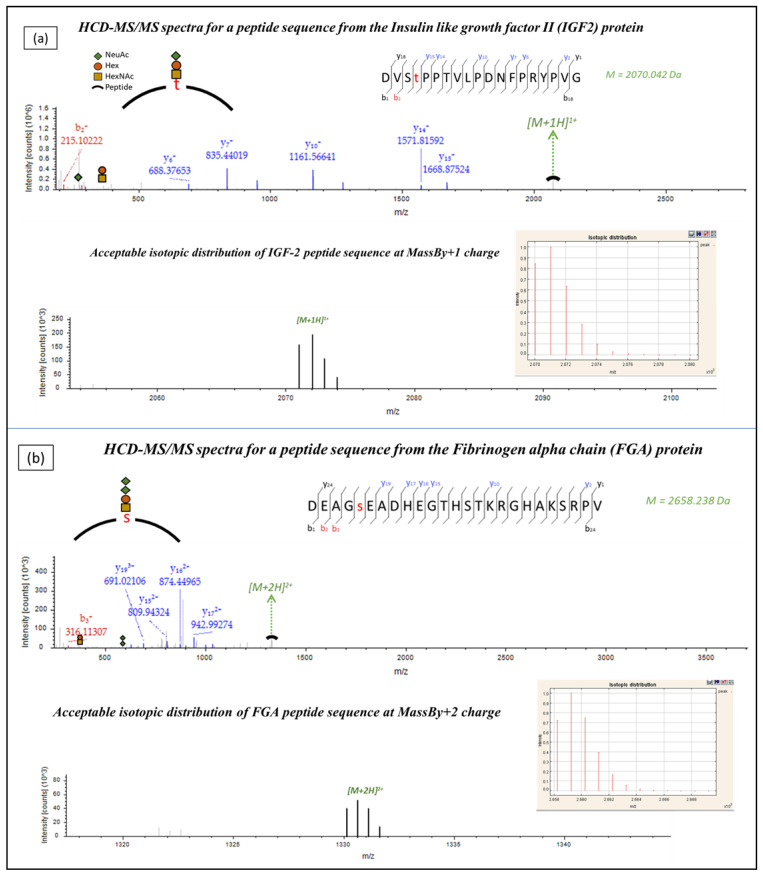
Quality check of HCD-MS/MS fragment spectra of the identified glycopeptides. Spectra as observed in Proteome Discoverer 1.4 are depicted along with their matched isotopic distributions of theoretical (insert) and experimental glycopeptide peaks. (**a**) Insulin-like growth factor II (IGF2) at a charge of +1 and (**b**) Fibrinogen alpha chain (FGA) at a charge of (+2).

**Figure 3 ijms-24-05402-f003:**
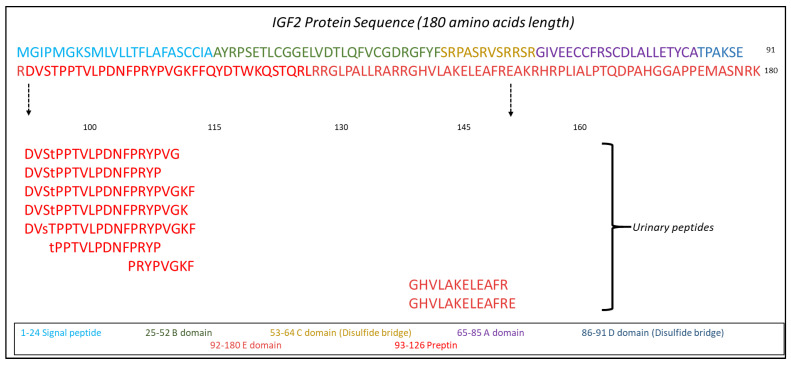
Identified naturally occurring IGF2 urinary peptides and their position in the IGF2 protein sequence. Lowercase “s” or “t” represent the presence of an O-linked glycan moiety.

**Figure 4 ijms-24-05402-f004:**
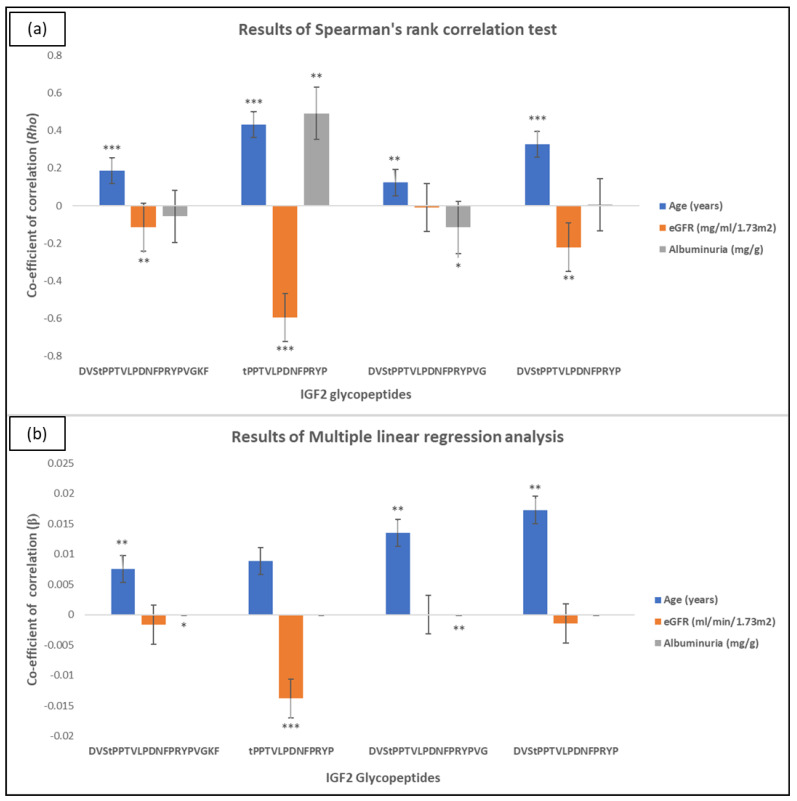
Results of (**a**) Spearman’s rank correlation test, (**b**) Multiple linear regression analysis for the abundance of four urinary IGF2 glycopeptides and Age, eGFR and Albuminuria. The statistically significant correlations are marked with asterisks (* *p* < 0.05, ** *p* < 0.0001, *** *p* < 0.00001).

**Figure 5 ijms-24-05402-f005:**
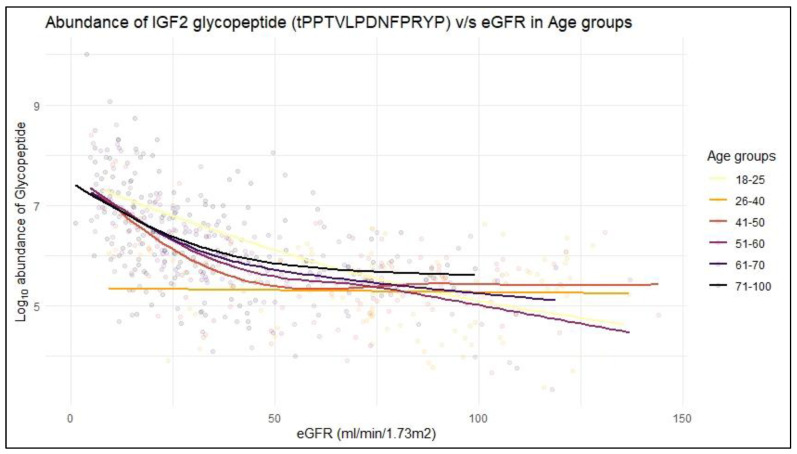
Correlation of the urinary IGF2 glycopeptide “tPPTVLPDNFPRYP” intensity with clinical parameter eGFR (mL/min/1.73 m^2^) for different age groups. Data were fit using a generalized additive model (GAM) suggesting significant correlations (*p* < 0.05) for all but one (age 61–70) age group.

**Figure 6 ijms-24-05402-f006:**
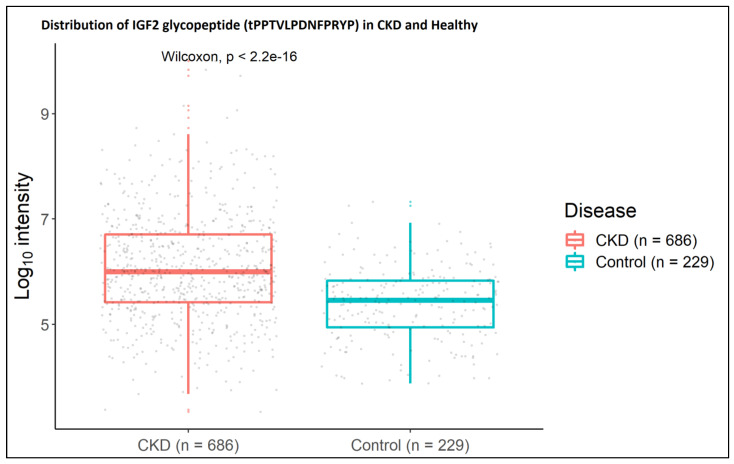
Box-Whisker plot depicting the increased urinary levels of IGF2 glycopeptide–tPPTVLPDNFPRYP in CKD subjects in comparison to healthy control subjects from the Human urinary proteome database.

**Figure 7 ijms-24-05402-f007:**
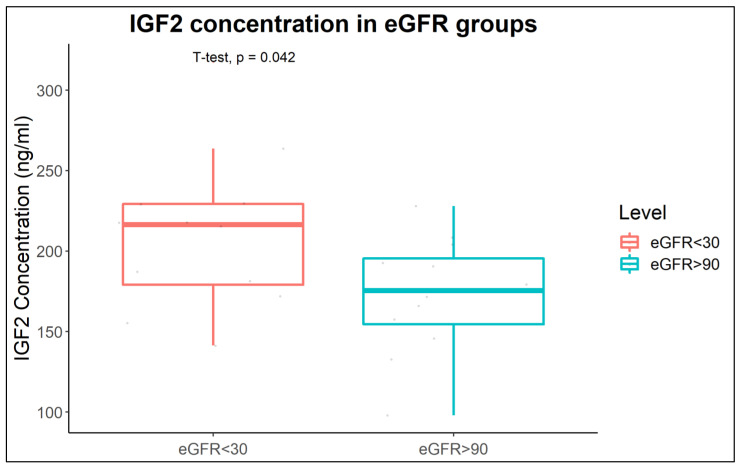
IGF2 protein levels in plasma of patients with eGFR < 30 mL/min/1.73 m^2^ in comparison to individuals with eGFR > 90 mL/min/1.73 m^2^ quantified by ELISA and depicted by a Box-Whisker plot.

**Table 1 ijms-24-05402-t001:** List of 17 O-linked glycopeptides identified by CE-MS/MS-based urinary glycopeptidomics analysis. The peptide frequency of detection (%) and abundance distribution in the Human urinary proteome database (*n* = 3810) are provided.

Glycopeptide Sequence	Mass	CE-Migration Time	Protein Name	Protein Accession ID	Gene Symbol	Glycan Composition	Glycan Position	Amino Acid		Frequency	Abundance	
	[Da]	[mins]		(UniProt)				Start	Stop	(%)	Range	Mean
DVStPPTVLPDNFPRYPVG	2726.28	42.55	Insulin-like growth factor II	P01344	IGF2	Hex(1)HexNAc(1)NeuAc(1)	T4	93	111	52.86	17.1–49,508.6	6115.74
tPPTVLPDNFPRYP	2269.05	39.25	Insulin-like growth factor II	P01344	IGF2	Hex(1)HexNAc(1)NeuAc(1)	T1	96	109	24.02	28.4–22,103.9	666.5
DVStPPTVLPDNFPRYP	2861.28	48.80	Insulin-like growth factor II	P01344	IGF2	Hex(1)HexNAc(1)NeuAc(2)	T4	93	109	23.62	21.4–27,681.8	2067.24
DVStPPTVLPDNFPRYPVGKF	3292.53	39.27	Insulin-like growth factor II	P01344	IGF2	Hex(1)HexNAc(1)NeuAc(2)	T4	93	113	89.74	19.0–85,843	9884.46
DVStPPTVLPDNFPRYPVGK	2854.37	35.74	Insulin-like growth factor II	P01344	IGF2	Hex(1)HexNAc(1)NeuAc(1)	T4	93	112	0.37	40.43–969.34	281.18
DVsTPPTVLPDNFPRYPVGKF	3001.44	33.45	Insulin-like growth factor II	P01344	IGF2	Hex(1)HexNAc(1)NeuAc(1)	S3	93	113	2.36	14.6–17,332.3	1904.29
tPAPLDSVFSSSERAHQVLR	3143.46	30.06	Vitamin K dependent protein C	P04070	PROC	Hex(1)HexNAc(1)NeuAc(2)	T1	19	39	1.58	10.0–589.32	185.93
tPAPLDSVFSSSERAHQ	2775.20	38.31	Vitamin K dependent protein C	P04070	PROC	Hex(1)HexNAc(1)NeuAc(2)	T1	19	39	11.05	6.56–4972.9	370.02
tPAPLDSVFSSSERAHQVLRI	3256.54	30.65	Vitamin K dependent protein C	P04070	PROC	Hex(1)HexNAc(1)NeuAc(2)	T1	19	40	37.09	16.78–35,415.4	2030.91
sALQPSPTHSSSNTQRLPDRVTG	3091.45	29.11	Inter-alpha-trypsin inhibitor heavy chain H1	P19827	ITIH1	Hex(1)HexNAc(1)NeuAc(1)	S1	645	668	0.05	50.69–97.77	74.23
LQPsPTHSSSNTQRLPD	2520.15	27.49	Inter-alpha-trypsin inhibitor heavy chain H1	P19827	ITIH1	Hex(1)HexNAc(1)NeuAc(1)	S4	648	664	0.55	12.12–270.39	117.54
LQPsPTHSSSNTQRLPDRVTG	2933.38	28.59	Inter-alpha-trypsin inhibitor heavy chain H1	P19827	ITIH1	Hex(1)HexNAc(1)NeuAc(1)	S4	648	668	0.32	44.7–3739.5	620.02
DEAGsEADHEGTHSTKRGHAKSRPV	3605.59	21.07	Fibrinogen alpha chain	P02671	FGA	Hex(1)HexNAc(1)NeuAc(2)	S5	605	629	20.71	18.9–4788.0	476.98
DEAGsEADHEGTHSTKRGHAKSRP	3215.42	19.996	Fibrinogen alpha chain	P02671	FGA	Hex(1)HexNAc(1)NeuAc(1)	S5	605	628	10.08	8.6–2873.3	266.59
DGGFDLSDALPDNENKKPtAIP	3260.45	38.15	CD99 antigen	P14209	CD99	Hex(1)HexNAc(1)NeuAc(2)	T19	23	44	25.51	13.6–14,136.9	986.31
tAVPSTGASKSKR	1944.95	24.88	Fibronectin	P02751	FN1	Hex(1)HexNAc(1)NeuAc(1)	T1	19	31	4.17	28.5–2130.5	425.66
ATIPRQDEVPQQtVAPQQQ	2789.31	40.96	Tumor necrosis factor receptor superfamily member 10D	Q9UBN6	TNFRSF10D	Hex(1)HexNAc(1)NeuAc(1)	T13	56	74	2.02	16.1–642.4	205.34

**Table 2 ijms-24-05402-t002:** Results of Spearman’s rank correlation test for the abundance of urinary IGF2 glycopeptides with eGFR (mL/min/1.73 m^2^) in different kidney disease etiologies. The significant correlations (*p* < 0.05) are marked with an asterisk.

Disease Etiology	IGF2 Glycopeptide							
	DVStPPTVLPDNFPRYPVGKF		tPPTVLPDNFPRYP		DVStPPTVLPDNFPRYPVG		DVStPPTVLPDNFPRYP	
	*Rho*	*n*	*Rho*	*n*	*Rho*	*n*	*Rho*	*n*
Healthy	−0.10 *	802	−0.23 *	117	−0.11	310	−0.21 *	175
ADPKD	−0.19 *	271	0.04	58	−0.05	220	−0.18 *	120
IgAN	−0.13 *	452	−0.51 *	188	0.08	237	−0.21	79
CKD	−0.32 *	577	−0.58 *	243	−0.04	415	−0.26 *	151
DKD	−0.28 *	198	−0.68 *	64	−0.01	132	−0.27	49
Nephrosclerosis	−0.18	118	−0.70 *	71	−0.05	68	−0.66	6
FSGS	−0.26 *	86	−0.69 *	32	0.01	57	−0.1	13
MCD	−0.41 *	52	−0.43	11	−0.25	29	−0.3	10
MGN	−0.31 *	72	−0.37	27	−0.19	49	−0.83 *	8
MPGN	−0.38	20	0.32	7	−0.04	16	0.4	4
Nephritis	−0.69 *	16	−0.6	9	0.17	9		
Tubullar Nephritis	−0.70 *	24	−0.86 *	18	−0.46	16		

CKD: Chronic kidney disease; IgAN: IgA nephropathy; ADPKD: Autosomal dominant polycystic kidney disease; DKD: Diabetic nephropathy; FSGS: Focal segmental glomerulosclerosis; MGN: Membranous glomerulonephritis; MCD: Minimal change disease; MPGN: Membranoproliferative glomerulonephritis.

**Table 3 ijms-24-05402-t003:** Clinical information of the study population. The mean values (ranges) are provided.

	Healthy Subjects (*n* = 2)	CKD Patients (*n* = 8)
Gender (% females)	100	37.5
Age (years)	29	38.5 (29–50)
MAP (mmHg)	<90	89.04 (77–111)
Proteinuria (g/day)	<0.1	1.28 (0.1–2.3)
eGFR (mL/min/1.73 m^2^)	>90	59.34 (13–99)

MAP: mean arterial blood pressure; eGFR: estimated glomerular filtration rate.

**Table 4 ijms-24-05402-t004:** Distribution of disease etiology, as obtained from the Human Urinary Proteome Database. (A small part, 391 subjects, were diagnosed with more than one kidney disease etiology).

Disease Etiology	Number of Datasets
Healthy controls	1616
Chronic kidney disease (CKD)	865
IgA nephropathy (IgAN)	586
Autosomal dominant polycystic kidney disease (ADPKD)	290
Diabetic nephropathy (DKD)	264
Nephrosclerosis	184
Focal segmental glomerulosclerosis (FSGS)	142
Membranous glomerulonephritis (MGN)	115
Minimal change disease (MCD)	64
Nephritis	33
Tubulointerstitial Nephritis	28
Membranoproliferative glomerulonephritis (MPGN)	28

**Table 5 ijms-24-05402-t005:** Clinical information of the study population. The mean values (ranges) are provided.

	eGFR > 90 mL/min/1.73 m^2^ (*n* = 12)	eGFR < 30 mL/min/1.73 m^2^ (*n* = 12)
Gender (% females)	16.7	75
Age (years)	38.1 (18–52)	46.3 (18–67)
SBP (mmHg)	120.3 (107–139)	135.8 (115–164)
DBP (mmHg)	77.91 (55–100)	84.4 (72–99)
eGFR (mL/min/1.73 m^2^)	107.8 (90–130)	23.2 (15–29)
Creatinine (mmol/day)	0.7 (0.5–0.9)	3.0 (1.8–4.1)
Hemoglobin (g/dL)	14.3 (12–16)	13.1 (11–16)
Cholesterol (mg/dL)	200.5 (155–252)	192.3 (142–253)

SBP: systolic blood pressure; DBP: diastolic blood pressure; eGFR: estimated glomerular filtration rate.

## Data Availability

The CE-MS/MS proteomics data have been deposited to the ProteomeXchange Consortium via the PRIDE partner repository with the dataset identifier PXD039829 and 10.6019/PXD039829.

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
