# Peer review of "Glycosylation Analysis of Urinary Peptidome Highlights IGF2 Glycopeptides in Association with CKD"

_ijms, 2023, doi:10.3390/ijms24065402_

Round 1

Reviewer 1 Report

The authors analyzed urine samples from patients with CKD and respective controls, in order to find urinary O-glycopeptides related to CKD. Their work is comprehensive and covers large set of information and data. However, the title and the aim of the study are not coordinated. In addition, the introduction is entirely based on the glycosylation and CKD, whereas there is no even peripheral information regarding IGF system components (peptides, binding proteins or receptors), their glycosylation or association with CKD.

As a reviewer, I found it difficult to follow the text, considering some of the data are experimentally obtained by the authors whereas others are drawn from the available repositories. This paper is more type of a meta-analysis/review and to my opinion, this should be clear for a reader. The difference between the original data and data collected for other authors should be more transparently stated.

Table 3 and 5 are missing information on the value given for age – it is not clear whether it is average or median or something else.

Author Response

“The authors analyzed urine samples from patients with CKD and respective controls, in order to find urinary O-glycopeptides related to CKD. Their work is comprehensive and covers large set of information and data.”

  • Thank you very much for this positive comment and acknowledging the efforts made behind the manuscript.

“However, the title and the aim of the study are not coordinated.”

  • This was a very interesting observation, for which we are grateful. We have changed the title of the manuscript to “Glycosylation analysis of urinary peptidome highlights IGF2 glycopeptides in association with CKD”, which we believe better describes the aim and findings of the manuscript.

“In addition, the introduction is entirely based on the glycosylation and CKD, whereas there is no even peripheral information regarding IGF system components (peptides, binding proteins or receptors), their glycosylation or association with CKD.”

  • This is a well taken point and we are sorry for the confusion.
  • The aim of our study was to identify naturally occurring urinary glycopeptides and their association with CKD. IGF2 glycopeptides were detected following application of our untargeted proteomics analysis. For this reason, we believe that we cannot add a reference to IGF2 in the introduction (as IGF2 was not the focus of the study from the beginning). Nevertheless, to address the well taken comment of the reviewer and avoid further potential confusion we changed the title of the manuscript to better reflect the presented workflow.

“As a reviewer, I found it difficult to follow the text, considering some of the data are experimentally obtained by the authors whereas others are drawn from the available repositories. This paper is more type of a meta-analysis/review and to my opinion, this should be clear for a reader. The difference between the original data and data collected for other authors should be more transparently stated.”

  • We apologize that our methodology was not clear enough and gave the impression this was a meta-analysis review.
  • The identification of the glycopeptides was conducted experimentally following profiling of urine samples (n=10) for their peptidome by CE-MS and subsequent glycosylation-focused mass spectra data analysis. For the further association of the identified peptides with CKD at high power, indeed an existing database was used (Human urinary proteome database) consisting of more than 20000 urinary peptide profiles representing multiple diseases (References in the manuscript for the database 21,29-30 and for the multiple studies 54-60). We have paid special attention to now make these points very clear, thus avoiding potential respective confusion to the reader.
  • A pilot experimental study was further conducted, targeting quantification of IGF2 protein in plasma (n=24) using ELISA (Line 191).

“Table 3 and 5 are missing information on the value given for age – it is not clear whether it is average or median or something else.”

  • This is a very well take point and a mistake on our behalf. The values that were provided in Tables 3 and 5 were the mean of the given values. We have now added this information to the table (along with the value range) and in addition, we changed the Table headings for better clarity.

Reviewer 2 Report

the  MS needs to be written in a scientific manner for publication, as it stands is more like a thesis. a lot of important figures/numbers are missing which must be in the MS to be able to reproduce the result. 

Author Response

“the MS needs to be written in a scientific manner for publication, as it stands is more like a thesis. a lot of important figures/numbers are missing which must be in the MS to be able to reproduce the result.”

  • We thank the reviewer for the comment, nevertheless as it was not specific enough it was difficult for us to understand which exactly figures the reviewer referred to. Nevertheless, we realized that some numbers (mean values-ranges) in Tables 3 and 5 were missing which have now been added. Furthermore, the manuscript was extensively revised for the language and we placed special attention to increase clarity so as to, among others, ensure data repeatability. Along these lines, the RAW data along with all the Proteome Discoverer output files are deposited in the ProteomeXchange consortium with the identifier being mentioned in the manuscript and the login details also provided in the cover letter.

In addition, we have provided all the protein and peptides identified per sample (as exported from the Proteome Discoverer) in the Supplementary Information (Table S1 and S2).